# Effectiveness of the ALT/AST ratio for predicting insulin resistance in a Korean population: A large-scale, cross-sectional cohort study

**Seul Ki Han**[1,2⊕], **Myung Jae Seo**[2,3⊕], **Taesic Lee**[2,3,4], **Moon Young Kim**[1,2]*

**1** Division of Gastroenterology and Hepatology, Department of Internal Medicine, Yonsei University Wonju College of Medicine, Wonju, Korea, **2** Regeneration Medicine Research Center, Yonsei University Wonju College of Medicine, Wonju, Korea, **3** Department of Family Medicine, Yonsei University Wonju College of Medicine, Wonju, Korea, **4** Division of Data Mining and Computational Biology, Institute of Global Health Care and Development, Wonju, Korea

⊕ These authors contributed equally to this work.
* drkimmy@yonsei.ac.kr

**Data Availability Statement:** The KNHANES is publicly available at https://knhanes.kdca.go.kr/-knhanes/eng/index.do.

## Abstract

Insulin resistance is a common pathophysiology in patients with type 2 diabetes mellitus, cardiovascular disease, and non-alcoholic fatty liver disease. Thus, screening for the risk of insulin resistance is important to prevent disease progression. We evaluated the alanine aminotransferase/aspartate aminotransferase (ALT/AST) ratio to predict insulin resistance in the general population, regardless of comorbidities. Datasets from the 2015, 2019, and 2020 Korea National Health and Nutrition Examination Surveys were used, and the following four indices were implemented to indicate insulin resistance: fasting serum glucose, insulin, homeostatic model assessment for insulin resistance (HOMA-IR), and β-cell function. We analyzed the degree of association between the liver enzyme profile and insulin resistance indices using Pearson's correlation coefficient and determined the associations using linear or logistic regression analysis. Accordingly, ALT levels in both sexes were positively and consistently correlated with the four aforementioned insulin resistance indices in stratification analyses based on diabetes, dyslipidemia, alcohol consumption, and obesity status. In multivariate linear regression, when comparing with ALT levels, the ALT/AST ratio exhibited superior predictive performance for fasting serum glucose and HOMA-β in Korean men and improved outcomes for all insulin resistance indices in Korean women. In this analysis that included a large community-based population, the ALT/AST ratio was a more useful predictive marker than the HOMA-IR. Regarding the predicted presence or absence of insulin resistance, the ALT/AST ratio could better predict HOMA-IR than the ALT level alone in Koreans. A simple, precise marker that represents the ALT/AST ratio could be a practical method to screen for insulin resistance in the general population, regardless of diabetes mellitus, alcohol intake, and sex.

**Funding:** The author(s) received no specific funding for this work.

**Competing interests:** The authors have declared that no competing interests exist.

## Introduction

With the increase in the prevalence of hypertension, dyslipidemia, type 2 diabetes mellitus (DM), and non-alcoholic liver disease (NAFLD), early detection of insulin resistance (IR) without specific symptoms is crucial. IR is a well-known risk factor for critical cardiovascular diseases (CVD). In individuals without DM, IR represented by homeostatic model assessment for insulin resistance (HOMA-IR), is an independent risk factor for stroke and coronary vascular disease [1–3]. Even in those with normal body weight and IR, the risk of incident DM and CVD events increases compared to the reference category [4]. According to a previous report, the estimated liver fat content significantly increased during follow-up in individuals with normal body weight and IR [4]. Similarly, reduced glucose metabolism has been reported in patients with NAFLD but without DM [5].

Therefore, a simple IR screening method is required. Sustained liver damage caused by IR leads to steatohepatitis and elevated liver enzyme levels. Considering the poor prognosis associated with elevated liver enzymes in NAFLD [6, 7], the liver enzyme profile can be used as a screening factor to detect IR, metabolic syndrome, or NAFLD. However, no screening methods or effective markers, including liver enzymes, have been suggested for the detection of IR in the general population by using public health checkup data. We aimed to determine indicators that could help detect IR using public health examination data.

## Materials and methods

### Dataset and participants

This study analyzed data from the Korea National Health and Nutrition Examination Survey (KNHANES) to confirm the association between the liver enzyme profile and IR. All the participants in the 2011–2019 KNHANES signed an informed consent form. The individual approval of the Institutional Review Board of Wonju Severance Christian Hospital was waived because the KNHANES data are publicly available and all participants in these surveys are fully anonymized and unidentified (IRB number: CR323312). The dataset was compiled from the official KNHANES website (https://knhanes.kdca.go.kr/knhanes/eng/index.do). This study contained detailed information about socioeconomic status; anthropometric measures; biochemical profiles, including fasting blood and urine; and nutrition patterns. Data on serum insulin, fasting blood glucose (Glc), and liver enzyme profile, such as ALT and AST levels, were obtained. Therefore, the 2015, 2019, and 2020 KNHANES sets, including laboratory indices for liver enzymes and IR, were analyzed in this study.

Study participants aged 20 years or older were selected. We excluded participants who tested positive for serological markers of viral hepatitis or were diagnosed with liver cirrhosis or hepatocellular carcinoma. Participants with complete lifestyle, medical, and laboratory data were included. A non-negligible number of outliers for ALT and AST levels and their ratios were observed (S1 Fig). An outlier value was defined as the mean ± three standard deviations (SD) of the liver enzyme levels. Therefore, participants showing outlier values for ALT level, AST level, or ALT/AST ratio were excluded. Finally, we selected 11,547 participants (men: 5,001, women: 6,546) to determine the association between the ALT/AST ratio and IR.

### Covariates

Hypertension was defined based on the seventh report of the Joint National Committee as follows: systolic blood pressure (BP) equal to or higher than 140 mmHg or diastolic BP equal to or higher than 90 mmHg; previous diagnosis of hypertension; or administration of antihypertensive medications [8]. DM was defined as a fasting Glc level ≥126 mg/dL; previous diagnosis

of DM; or administration of anti-glycemic drugs. Regarding dyslipidemia status, the KNHANES contains both questionnaire- and laboratory-based data. However, several participants had missing high-density lipoprotein cholesterol and low-density lipoprotein cholesterol values. Therefore, we depended only on questionnaire-based information and defined dyslipidemia based on the administration of antilipidemic medications.

Several criteria for regular alcohol consumption have been reported [9, 10]. Lee et al. [9] defined routine alcohol consumption as that of more than 140 g/week in men and more than 70 g/week in women. In another study by Kang et al. [10], participants who consumed more than 210 g/week (men) or more than 140 g/week (women) of alcohol were categorized into the regular alcohol consumption group. Considering these studies and the distribution of the number of alcohol consumption groups, the regular alcohol consumption group was determined based on a sex-specific cut-off (men and women consuming more than 140 g/week and more than 70 g/week, respectively).

Several studies have used obesity status as a covariate [10, 11]. One study defined obesity by body mass index (BMI) [10] and another determined obesity by waist circumference (WC) [11], both of which are highly correlated [12, 13]. Recently, several terms, including the obesity paradox, normal-weight obesity, and central obesity, have been introduced, triggering the need for alternative BMI indices for evaluating body composition and obesity. Thus, we included both BMI and WC as covariates [14, 15]. Based on the Korea Society for the Study of Obesity Guideline [16], the obesity group was characterized as follows: general obesity when BMI was equal to more than 25 and central obesity when the WC was equal to or more than 90 cm in men and 85 cm in women.

Homeostasis model assessment for insulin resistance (HOMA-IR) and HOMA for β-cell (HOMA-β) were calculated using the following equations: HOMA-IR is calculated by multiplying serum insulin by fasting blood glucose, then dividing by the constant 405; HOMA-β is calculated by multiplying the constant 360 by serum insulin, then dividing by the value of fasting blood glucose minus 63 [17].

## Statistics

We implemented a one-way analysis of variance to test for the linear differential characteristics of the covariates according to the tertile groups for the ALT/AST ratio. The mean values of each tertile group for the ALT/AST ratio were defined as representative values for the three groups in the form of continuous variables. For categorical variables, the chi-square test was used to analyze the differences among tertiles for ALT/AST ratio. A comparative analysis of the two continuous variables (serum ALT vs. Glc) was conducted using Pearson's correlation coefficient (PCC). A univariate (Model 1) or multivariate linear (Models 2–4) regression (LiR) models were implemented to determine the association between liver enzyme profile (i.e., ALT, AST, and ALT/AST ratio) and IR index (i.e., Glc, insulin, HOMA-IR, and HOMA-β). We set the liver enzyme profiles and each IR index as the dependent and independent variables, respectively. Three different multivariate models (Models 2–4) were used to determine the independent relationship between liver enzyme profile and IR indices according to four combinations of confounders: Model 2 (age), Model 3 (age, DM, dyslipidemia, and alcohol consumption), and Model 4 (age, DM, dyslipidemia, alcohol, BMI, and WC).

Classification performance was evaluated based on the area under the curve (AUC) of the receiver operating characteristic (ROC). All statistical analyses were performed using R software (version 4.1.2). We set a two-tailed p-value less than 0.05 as the significance level.

## Results

### General characteristics

**Table 1** presents the sex-specific medical laboratory characteristics according to tertiles for the ALT/AST ratio. The ALT/AST ratios were sorted in ascending order and categorized into three groups (T1, T2, and T3) based on tertiles. With an increase in the ALT/AST ratio in Korean men, the following characteristics showed a higher trend: ALT level, AST level, younger age, hypertension, DM, dyslipidemia, alcohol consumption, BMI, WC, insulin levels, Glc levels, HOMA-IR, and HOMA-β (**Table 1**). In Korean women, most features, except for age, exhibited a similar higher trend related to increasing ALT/AST ratio as Korean men. The age of Korean women was positively correlated with the ALT/AST ratio (**Table 1**).

### Correlational analyses of the liver enzyme profile with IR indices

Comparative analyses were conducted based on PCC to evaluate the degree of association between liver enzyme profile and IR indices. In Korean men, serum ALT level was positively correlated with the four IR indices, including Glc, insulin, HOMA-IR, and HOMA-β. These positive relationships persisted after selecting participants diagnosed with DM or dyslipidemia; who consumed alcohol; or those with general or abdominal obesity (**Fig 1A**). In Korean men, serum AST levels were correlated with most IR indices, except for Glc in patients with DM and HOMA-β in patients with DM or dyslipidemia (**Fig 1A**). In Korean men, the ALT/AST ratio was significantly associated with the four IR indices in all participants and all stratification analyses (**Fig 1A**). In Korean women, the positive association of the ALT/AST ratio with

**Table 1. Sex-specific general characteristics according to the tertile of alanine aminotransferase/aspartate aminotransferase ratio.**

|  | Korean men | | | | Korean women | | | |
|---|---|---|---|---|---|---|---|---|
|  | T1 | T2 | T3 | *P* for trend | T1 | T2 | T3 | *P* for trend |
| N | 1639 | 1690 | 1672 |  | 2136 | 2199 | 2211 |  |
| ALT, mg/dL | 15.9 ± 0.14 | 22.8 ± 0.19 | 35.9 ± 0.19 | <0.001 | 12.6 ± 0.08 | 17.3 ± 0.1 | 26.1 ± 0.1 | <0.001 |
| AST, mg/dL | 24.9 ± 0.2 | 24.6 ± 0.2 | 26.4 ± 0.2 | <0.001 | 22.2 ± 0.13 | 22.2 ± 0.13 | 23.7 ± 0.12 | <0.001 |
| ALT/AST ratio | 0.644 ± 0.0029 | 0.923 ± 0.0018 | 1.352 ± 0.0018 | <0.001 | 0.57 ± 0.002 | 0.779 ± 0.0012 | 1.098 ± 0.0012 | <0.001 |
| Age, years | 64.1 ± 0.28 | 60 ± 0.27 | 55.3 ± 0.27 | <0.001 | 60.3 ± 0.28 | 59.8 ± 0.24 | 58.4 ± 0.24 | <0.001 |
| HTN, n | 763 (46.6) | 780 (46.2) | 799 (47.8) | 0.614 | 765 (35.8) | 877 (39.9) | 987 (44.6) | <0.001 |
| DM, n | 273 (16.7) | 329 (19.5) | 390 (23.3) | <0.001 | 212 (9.9) | 248 (11.3) | 444 (20.1) | <0.001 |
| ALM, n | 212 (12.9) | 262 (15.5) | 339 (20.3) | <0.001 | 334 (15.6) | 453 (20.6) | 594 (26.9) | <0.001 |
| Alcohol consumption, n | 1131 (69.0) | 1179 (69.8) | 1260 (75.4) | <0.001 | 2002 (93.7) | 2031 (92.4) | 2080 (94.1) | 0.054 |
| BMI, kg/m² | 23.2 ± 0.07 | 24.4 ± 0.07 | 25.8 ± 0.07 | <0.001 | 22.8 ± 0.06 | 23.7 ± 0.07 | 25.1 ± 0.07 | <0.001 |
| Obesity based on BMI, n | 414 (25.3) | 658 (38.9) | 973 (58.2) | <0.001 | 468 (21.9) | 647 (29.4) | 1042 (47.1) | <0.001 |
| WC, cm | 85.2 ± 0.21 | 88.3 ± 0.19 | 91.6 ± 0.2 | <0.001 | 79.5 ± 0.19 | 81.8 ± 0.19 | 85.3 ± 0.19 | <0.001 |
| Obesity based on WC, n | 468 (28.6) | 672 (39.8) | 957 (57.2) | <0.001 | 556 (26.0) | 756 (34.4) | 1109 (50.2) | <0.001 |
| Insulin, uIU/mL | 6.2 ± 0.1 | 7.7 ± 0.11 | 10.5 ± 0.11 | <0.001 | 6.6 ± 0.09 | 7.5 ± 0.09 | 10.1 ± 0.09 | <0.001 |
| Glc, mg/dL | 104 ± 0.56 | 106.8 ± 0.62 | 111.3 ± 0.63 | <0.001 | 97.1 ± 0.38 | 99.8 ± 0.4 | 107.3 ± 0.4 | <0.001 |
| HOMA-IR | 1.7 ± 0.03 | 2.1 ± 0.03 | 3 ± 0.03 | <0.001 | 1.6 ± 0.03 | 1.9 ± 0.03 | 2.8 ± 0.03 | <0.001 |
| HOMA-β | 60 ± 1.44 | 72 ± 1.23 | 92.5 ± 1.24 | <0.001 | 72.6 ± 1.69 | 78.9 ± 1.51 | 94.7 ± 1.51 | <0.001 |

All data in the KNHANES are presented as mean ± standard error for continuous variables and as frequency and percentage (%) for categorical variables. The ALT/AST ratios were sorted in ascending order and categorized into three groups (T1, T2, and T3) based on tertiles. Abbreviations: KNHANES, Korea National Health and Nutrition Examination Survey; ALT, alanine aminotransferase; AST, aspartate aminotransferase; HTN, hypertension; DM, diabetes mellitus; ALM, anti-lipidemic medication; BMI, body mass index; WC, waist circumference; Glc, fasting blood glucose; HOMA-IR, Homeostasis model assessment for insulin resistance; HOMA-β, HOMA for β cell.

several IR indices was most remarkable concerning the liver enzyme profile (ALT, AST, and ALT/AST ratio, **Fig 1B**).

## Comparative analysis of the association of the liver enzyme profile (ALT/ AST ratio vs. ALT level) with IR indices

Four LiR models, including univariate and three multivariate models with different combinations of covariates, were implemented to identify associational trends (i.e., beta coefficients) between the liver enzyme profile and IR indices and their significance (*p*-value). In Korean men model 1 (univariate model), the ALT/AST ratio showed higher beta-coefficients and matched statistics for Glc and HOMA-β than for ALT level (**Fig 2**). In model 2, the beta coefficients and matched significant powers of the ALT/AST ratio were higher for all four IR indices than for ALT level (**Fig 2**). In model 4 including age, DM, dyslipidemia, alcohol consumption, BMI, and WC as covariates, the beta coefficients and matched significant powers of the ALT/ AST ratio were higher than that of ALT level only for the IR indices of Glc and HOMA-β (**Fig 2**).

In Korean women, all models demonstrated that the associational trends of the ALT/AST ratio (i.e., beta coefficients) and matched significant powers (i.e.,–log[*p*-value]) were higher than that of ALT level for all four IR indices (**Fig 3**). To minimize the effect of the scale of variables on the beta coefficient, which quantifies the association between two variables, both the dependent and independent variables were transformed into z-scores.

## Comparative analysis of the classification performances of the liver enzyme profile (ALT/AST ratio vs. ALT level) for IR status

To transform the four continuous IR levels into binomial features (i.e., presence or absence of IR status), we operationally defined participants meeting the following criteria as having IR: Glc level higher than 126 mg/dL, serum insulin higher than or equal to the mean plus one SD of serum insulin, HOMA-IR higher than or equal to the mean plus one SD of HOMA-IR, and HOMA-β higher than or equal to the mean plus one SD of β-cell function (**Fig 4**). We then evaluated the classification performance of the ALT/AST ratio or ALT level for the four binomial IR statuses based on true and false positive rates and their combinatory index, referred to as AUC. The performance of the ALT/AST ratio in classifying IR status was comparable to that of the ALT level (**Fig 4**). In Korean men, the best performance was achieved when classifying IR status defined by serum insulin using serum ALT levels (**Fig 4**). In Korean women, the best AUC value was obtained when predicting IR status based on HOMA-IR using the ALT/ AST ratio (**Fig 4**). In Korean men, adults (age: 40–60 years), older participants (aged more than 60 years), and those consuming alcohol showed similar performance of ALT level and ALT/AST ratio for predicting IR status as all Korean men. Moreover, among Korean women, adults (aged 40–60 years) and the alcohol consumption groups exhibited higher AUC levels than all participants. Moreover, these groups performed better for most IR states using the ALT/AST ratio than the individual ALT levels.

## Discussion

Using nationwide Korean datasets, we demonstrated the relationship between ALT level and the ALT/AST ratio with four IR indices. A positive association between the liver enzyme profile and IR was common in both Korean men and women. Moreover, the association between the hepatic steatosis indices and IR remained independently significant after adjusting for

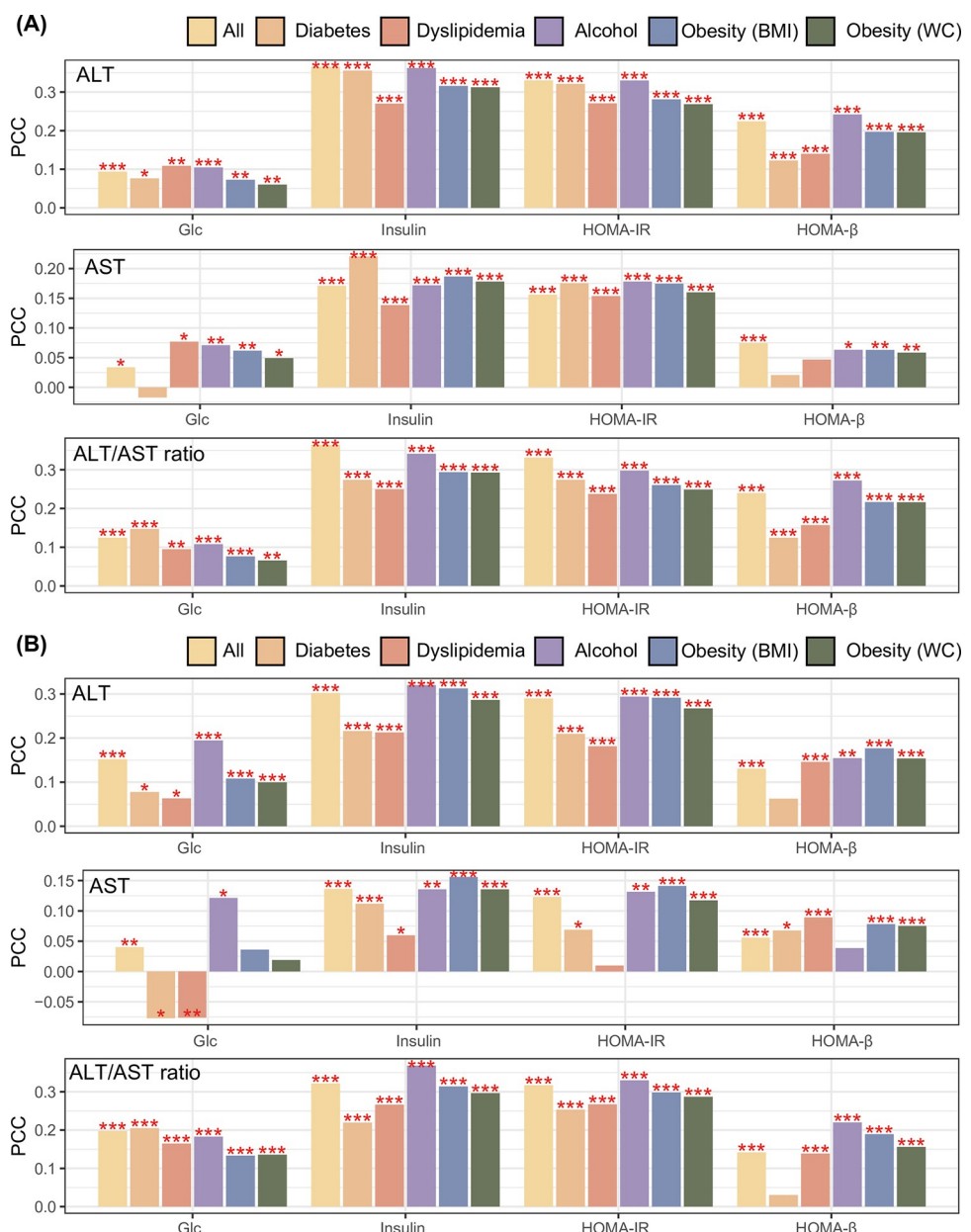

**Fig 1.** Association between the liver enzyme profile and IR indices in (A) Korean men and (B) Korean women. *, **, and *** denote *p*-value < 0.05, *p* < 0.01, and *p* < 0.001, respectively, calculated by Pearson's correlation method. Pearson's correlation was used to identify the association of liver profiles, including ALT, AST, and their ratio, with the four IR indices. Correlation analysis was repeated six times with different combinations of participants (those with diabetes, dyslipidemia, alcohol consumption, general obesity, and abdominal obesity). Abbreviations: IR, insulin resistance; PCC, Pearson's correlation coefficient; ALT, alanine aminotransferase; AST, aspartate aminotransferase; Glc, fasting blood glucose; HOMA-IR, Homeostasis model assessment for insulin resistance; HOMA-β, HOMA for β-cells.

various clinical variables. Stratification analyses in various settings revealed that the ALT/AST ratio was more reflective of IR status than the ALT level.

The definitive criteria of normal serum ALT level vary according to sex, race, region, age, and even institution [18–20]. However, serum ALT levels increase linearly with liver-related

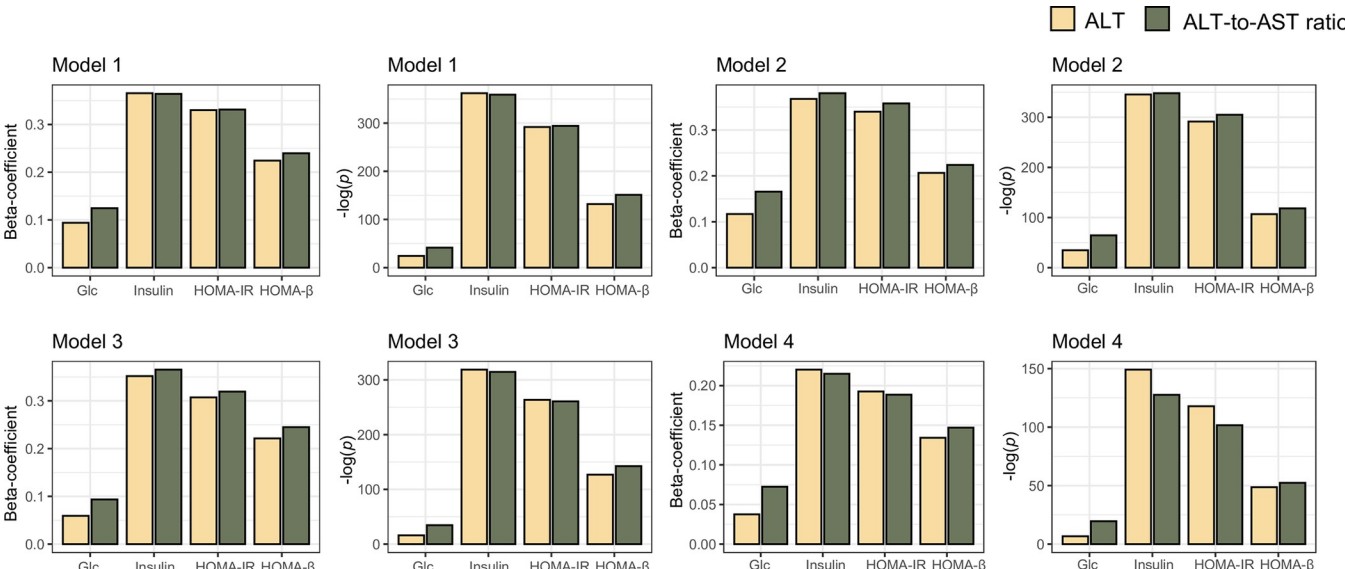

**Fig 2. Comparative analysis of the degree of association of ALT and ALT/AST ratio with IR indices in Korean men.** The association between the liver enzyme profile and IR index was calculated using four linear regression models, including different combinations of covariates. Model 1 was a univariate model with the liver enzyme profile and IR indices as the dependent and independent variables, respectively. Model 2 included age as the covariate. Model 3 included age, DM, dyslipidemia, and alcohol consumption as confounders. Model 4 included age, DM, dyslipidemia, alcohol consumption, BMI, and WC as confounding factors. Abbreviations: ALT, alanine aminotransferase; AST, aspartate aminotransferase; HTN, hypertension; DM, diabetes mellitus; BMI, body mass index; WC, waist circumference; Glc, fasting blood glucose; HOMA-IR, Homeostasis model assessment for insulin resistance; HOMA-β, HOMA for β cell.

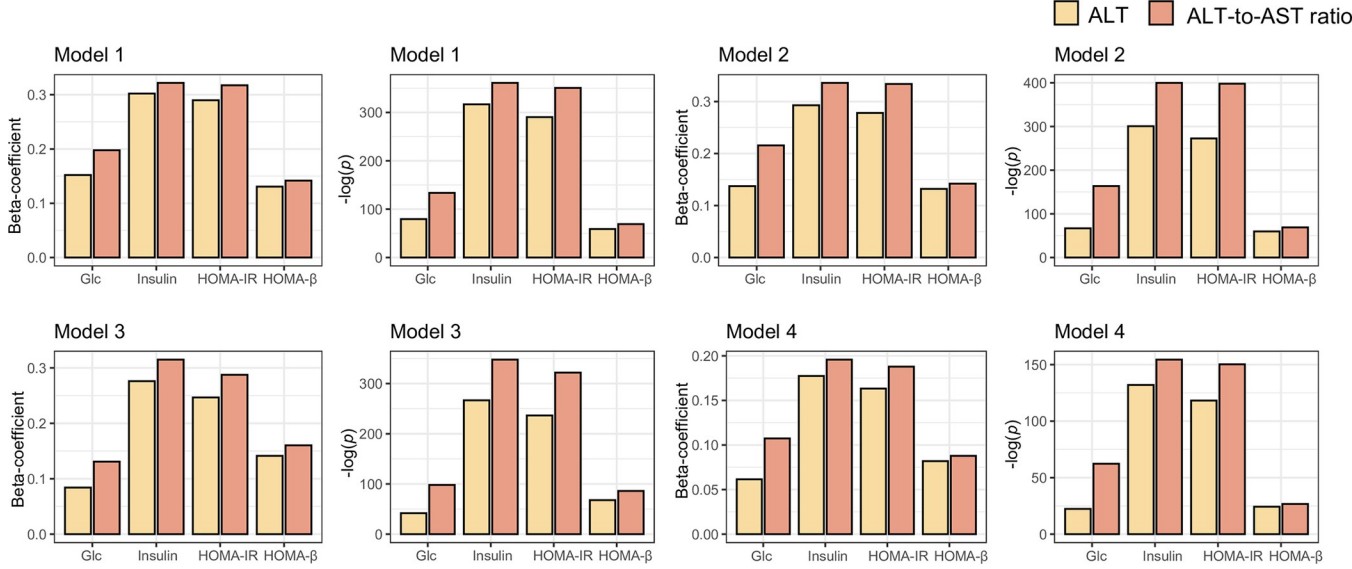

**Fig 3. Comparative analysis of the degree of association of ALT and ALT/AST ratio with IR indices in Korean women.** The association between the liver enzyme profile and IR index was calculated using four linear regression models including different combinations of covariates. Model 1 was a univariate model setting the liver enzyme profile and IR indices as the dependent and independent variables, respectively. Model 2 included age as the covariate. Model 3 included age, DM, dyslipidemia, and alcohol consumption as confounders. Model 4 included age, DM, dyslipidemia, alcohol consumption, BMI, and WC as confounding factors. Abbreviation: ALT, alanine aminotransferase; AST, aspartate aminotransferase; HTN, hypertension; DM, diabetes mellitus; BMI, body mass index; WC, waist circumference; Glc, fasting blood glucose; HOMA-IR, Homeostasis model assessment for insulin resistance; HOMA-β, HOMA for β cell.

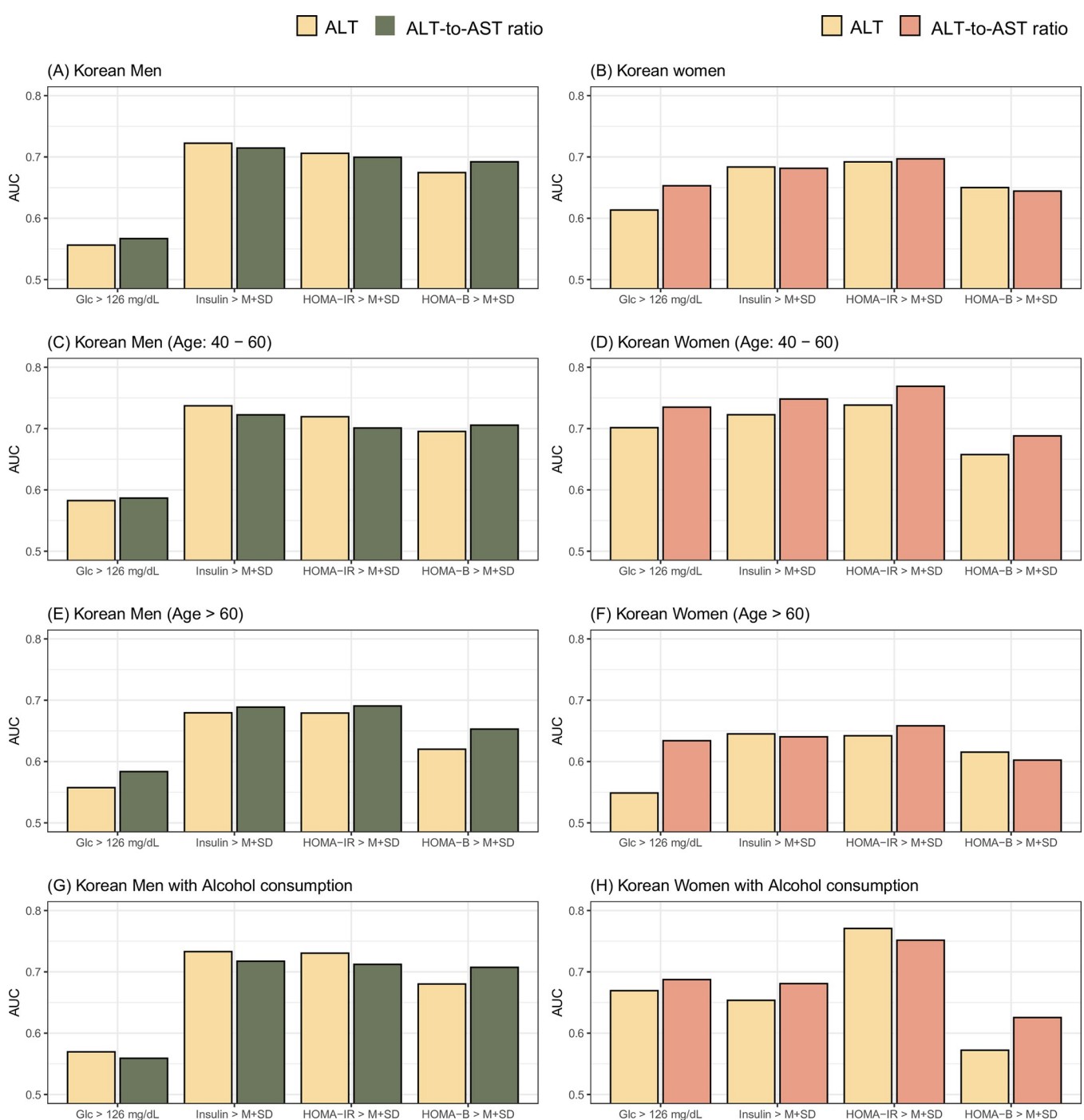

**Fig 4.** Classification performance of ALT and ALT/AST ratio for IR status in (A) Korean men and (B) Korean women. Abbreviations: ALT, alanine aminotransferase; AST, aspartate aminotransferase; IR, insulin resistance; AUC, area under curve of receiver operating characteristic; Glc, fasting blood glucose; HOMA-IR, Homeostasis model assessment for insulin resistance; HOMA-β, HOMA for β cell; M+SD; mean plus one standard deviation.

mortality even at relatively low serum levels [21]. In previous studies that included large numbers of healthy individuals, elevated serum aminotransferase levels, even those within the normal range, were associated with liver-related mortality and metabolic syndrome [22–24]. Considering that some patients with non-alcoholic steatohepatitis present with normal serum

ALT levels [25, 26], lower ALT levels should be monitored for the risk of progression of liver disease and other metabolic disorders.

Generally, increased AST or ALT levels are considered liver injury markers. However, some studies have reported that amino acid transferases may be associated with glucose metabolism and metabolic syndromes [27, 28]. Through the action of ALT, glutamate can transfer its amino acid group to pyruvate, forming alpha-ketoglutarate and alanine [29]. Glutamate likely stimulates glucagon release from the pancreas and increases the transamination of pyruvate to alanine, which strongly promotes gluconeogenesis in obese individuals. The hypothesis that the glutamate-glutamine cycle is related to DM and glucose intolerance has been suggested in clinical and preclinical studies [30, 31]. This concept of metabolism raises new questions regarding the role of elevated aminotransferase levels in metabolic syndrome. Additionally, our results support the hypothesis that elevated aminotransferase levels are related to glucose metabolism, especially in relation to HOMA-IR.

HOMA-IR indicates the presence of IR and is a predictor of diabetes. However, to calculate the HOMA-IR, additional blood sampling and fasting time are required. Therefore, more effective screening methods are required. The ALT/AST ratio as a tool for predicting IR has been suggested previously [32]. Kawamoto et al. also reported that the ALT/AST ratio was the best surrogate marker for IR compared to other markers, such as serum triglyceride, gamma-glutamyltransferase, ALT, or AST levels alone [33]. This finding is supported by another study results [34]. Recently, a relationship between the ALT/AST ratio and diabetes and prediabetes stages has also been reported in China [35]. The current study is the first to include more than 10,000 Korean participants. Moreover, the predictive power was better in middle-aged women and individuals who consumed alcohol (**Fig 4**). Based on these results, in the group corresponding to T2 or T3, the prevalence of diabetes, hypertension, and HOMA-IR significantly increased. This means that metabolic syndrome management is required in the T2 and T3 groups.

Men and women differed substantially regarding IR, body composition energy balance, and metabolic composition [36–39]. A previous study reported age- and sex-specific differences in HOMA-IR levels, with increased levels in women [40]. Moreover, fat distribution in the visceral, hepatic, peripheral, or subcutaneous adipose tissue was presumed to contribute to this phenomenon [38, 41]. In our study, the predictive value of the ALT/AST ratio compared to HOMA-IR was superior in women than in men. Presumably, corrected AST levels, muscle mass, and alcohol intake may have influenced the results.

In patients with DM, the relationship between the ALT/AST ratio and IR was weaker than that in the other comorbidity groups. These patients were likely to take oral antidiabetic drugs to improve their IR. Metformin, which is recommended as a primary medication for type 2 DM, is known to improve IR [42–44]. Therefore, the predictive value of ALT/AST markers in patients with diabetes tended to be low in our analysis.

Our study has several strengths. First, it included a large Korean population. A recent study validated the association between ALT/AST ratio and IR in a female Japanese population of approximately 460 individuals, whereas our study demonstrated an association between liver enzyme levels and IR in more than 10000 Korean men and women, which may be more generalizable [45]. Second, various analyses, including stratification and predictive value analyses, confirmed an independent relationship between ALT/AST ratio and IR. Although the earlier research result is similar to ours in comparing the relevance of various IR indices to the ALT/AST ratio [45], our series of analyses allowed us to explore this relationship from a broader perspective and revealed the novel finding that ALT/AST has better predictive power than the ALT level.

This study had some limitations. First, we collected only serum aminotransferase levels without any radiological data. Whereas the detection of steatosis could have supported our

study hypothesis, radiological data were not included in the analysis. Second, aminotransferase is a sensitive marker of liver dysfunction; however, its specificity is low because it can be elevated under other conditions. We intended to strengthen the specificity by estimating the ALT/AST ratio. Third, this was a cross-sectional study. Follow-up studies on clinical outcomes, such as the development of metabolic syndrome or type 2 DM, are needed to reinforce its prognostic significance. Further studies on the exact mechanisms and related gene expressions are required. Finally, owing to the retrospective, cross-sectional nature of the study, there were some limitations. For example, in diagnosing diabetes, it is accurate to measure fasting blood glucose once and then examine the two results. However, the KNHANES only includes a single-point fasting glucose level; therefore, DM was defined based on a single-point glucose level. As many confounding factors are implicated in the pathogenetic relationship with the liver, the evaluation of multi-pathogenetic interactions would take time.

## Conclusions

Our findings suggest that liver transaminases, especially the ALT/AST ratio, could be considered a biomarker of liver damage and a phenotype of metabolic syndrome. This study suggests a practical method to screen for IR in a large general population, including healthy individuals, regardless of sex and alcoholism.

## Supporting information

**S1 Fig.** Distributions of ALT, AST, ALT/AST in Korean Men (A) and women.
(DOCX)

**S1 File.**
(ZIP)

## Author Contributions

**Conceptualization:** Seul Ki Han, Taesic Lee, Moon Young Kim.

**Data curation:** Seul Ki Han, Taesic Lee.

**Formal analysis:** Seul Ki Han, Taesic Lee, Moon Young Kim.

**Methodology:** Seul Ki Han, Myung Jae Seo, Taesic Lee, Moon Young Kim.

**Software:** Taesic Lee.

**Supervision:** Moon Young Kim.

**Visualization:** Myung Jae Seo, Taesic Lee.

**Writing – original draft:** Seul Ki Han, Myung Jae Seo, Taesic Lee.

**Writing – review & editing:** Myung Jae Seo, Moon Young Kim.

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
