## [Decision Letter · Decision Letter 0]

22 Jan 2024

PONE-D-23-35138ALT/AST ratio: the useful predictive marker for insulin resistancePLOS ONE

Dear Dr. Kim,

Thank you for submitting your manuscript to PLOS ONE. After careful consideration, we feel that it has merit but does not fully meet PLOS ONE’s publication criteria as it currently stands. Therefore, we invite you to submit a revised version of the manuscript that addresses the points raised during the review process.

**Before considering the paper for potential publication, authors are recommended to timely and carefully address all the reviewers' comments.**

We look forward to receiving your revised manuscript.

Kind regards,

Anna Di Sessa, PhD, MD

Academic Editor

PLOS ONE

Journal Requirements:

3. We note that your Data Availability Statement is currently as follows: The KNHANES is publicly available at https://knhanes.kdca.go.kr/-knhanes/eng/index.do.

4. Please amend your authorship list in your manuscript file to include author Dr. Myung Jae Seo.

Reviewers' comments:

Reviewer's Responses to Questions

**Comments to the Author**

1. Is the manuscript technically sound, and do the data support the conclusions?

Reviewer #1: No

Reviewer #2: Partly

Reviewer #3: Yes

Reviewer #4: Yes

2. Has the statistical analysis been performed appropriately and rigorously? 

Reviewer #1: No

Reviewer #2: N/A

Reviewer #3: Yes

Reviewer #4: Yes

3. Have the authors made all data underlying the findings in their manuscript fully available?

Reviewer #1: No

Reviewer #2: No

Reviewer #3: Yes

Reviewer #4: No

4. Is the manuscript presented in an intelligible fashion and written in standard English?

Reviewer #1: No

Reviewer #2: Yes

Reviewer #3: Yes

Reviewer #4: Yes

5. Review Comments to the Author

Reviewer #1: Title: "ALT/AST ratio: the useful predictive marker for insulin resistance".

Keywords : Keywords should include AST and ALT, ALT/AST

Abstract: Why are there 2 different abstracts?

Abstract- Background: In one of the abstracts (Alanine aninotransferase/Aspartate aminotransferase ratio for analysis.), the abbreviation ALT/AST was not written.

Abstract- Results: Only the findings should be included in the results section. Comments should be avoided. Findings should be presented clearly and statistically in the form of group comparisons.

1. Introduction:

a- Why is NAFLD described as dominant rather than insulin resistance and homo-IR? NAFLD relationship is not written in the conclusion section. If NAFLD is an primary issue, this should be stated in the study title.

b- Introduction and conclusion sections are unrelated.

c- alanine aminotransaminase protein (ALT). protein ??

2. Methods

a- 2.1.Dataset and Participants. The patient population should be clearly differentiated. This section should be summarized. If NAFLD is present, it should be stated whether USG scanning was performed or not.

b- Diabetes mellitus was defined as serum fasting glucose level ≥126 mg/dL; previous diagnosis; anti-glycemic drugs. Has this been confirmed 2 times ?

c- (high- and low-density lipoprotein (HDL and LDL) cholesterol.) HDL- kolesterol ? LDL-kolesterol ?

3.Results:

a- ALT-to-AST ratio. ALT/ALT. please make a standard abbreviation

b- Please write the long version of the abbreviations in table 1 in the description.

c- What are the criteria for T1-2-3 groups

d- What are the r and p values for the ALT/AST correlation with HOMA-IR? ?

4. Dissuscion: Findings should be discussed clearly.

Reviewer #2: 1-İn the Method/data set and participants

Last sentence ^^ Finally, we selected a population of 11,547 individuals (men: 5,001, women: 6,546) to determine the association between ALT/AST ratio and NAFLD.

However, to determine the association of ALT/AST ratio with NAFLD, there must be a definite indication of the presence of NAFLD in the participants, such as abdominal ultrasonography or liver biopsy, and the association must be analysed accordingly. Otherwise, such a result cannot be reported.

2-While the relationship with diabetes, dyslipidaemia and obesity is defined in Figure 1, was the relationship with the healthy control group analysed?

2-İn the Table 1: ALT/AST ratio analysed according to tertile groups; the definition of tertile groups should be better explained.

3-İn the Table 1: ALT/AST ratio analysed according to tertile groups; tertile groups should be better explained. 3-Table 2: ALT/AST ratio analysed by tertile groups; tertile groups need to be better explained. This situation prevents the evaluation of the relationship between ALT/AST ratio and insulin resistance independently of liver diseases.

4- There are many studies investigating the relationship between ALT/AST ratio and insulin resistance in obese, healthy individuals, NAFLD, etc. in the literature. At this point, I recommend hypothesising by carefully reading the literature in the selection of method material.

Reviewer #3: This work is important and well-structured and the results support the authors ‘conclusions. Statistical procedures seemed steady. Only, the following minor comments should be considered.

1. The type of study is better to be mentioned in the title.

2. Please clarify statistical analysis section in more detail. For example, method of controlling confounders in this study.

3. Authors need to add the strengths of this study before the paragraph explaining study limitations.

4. Please check the concordance of the tenses used throughout the text.

5. Please define all nonstandard abbreviations used in the text and abstract.

Reviewer #4: In this work the authors present a statistical study on predicting insulin resistance with the AST/ALT. However, the presented work is very similar with “Associations of alanine aminotransferase/aspartate aminotransferase with insulin resistance and β-cell function in women”, https://doi.org/10.1038/s41598-023-35001-1. Unless the authors can demonstrate that their work is significantly different from the published work, I do not recommend the paper to be published.

6. PLOS authors have the option to publish the peer review history of their article (what does this mean?). If published, this will include your full peer review and any attached files.

Reviewer #1: No

Reviewer #2: No

Reviewer #3: No

Reviewer #4: No

---

## [Author Response · Author response to Decision Letter 0]

7 Mar 2024

Professor Emily Chenette

Editor-in-Chief

PLOS ONE

Dear Professor Emily Chenette.

Manuscript Number: PONE-D-23-35138

Manuscript Title: Effectiveness of ALT/AST ratio for predicting insulin resistance in Korean population: Large-scale, cross sectional cohort study

On behalf of all the authors, I would like to express our sincere gratitude for giving your thorough consideration and scrutiny to the manuscript entitled “Effectiveness of ALT/AST ratio for predicting insulin resistance in Korean population: Large-scale, cross sectional cohort study”. As the reviewer pointed out, we revised the text and changed the title. The accurate and keen comments made by the Reviewers have allowed us to embrace the critical issues in this paper. We did our best to attain the scientific and literary standard required by the Reviewers and we have revised the manuscript according to the Reviewers’ suggestions. The revised manuscript is the result of our hard work and we sincerely hope that our revision will be considered positively during further review. We are aware of the importance of the scrutinizing efforts made by the Reviewers on the scientific and clinical merit of our manuscript. The changes in the revised manuscript have been highlighted (in red). Specific responses to Editor’s and Reviewers’ comments are included below.

Answers to comments of Reviewer #1:

1. Abstract: Why are there 2 different abstracts? Abstract- Background: In one of the abstracts (Alanine aminotransferase/Aspartate aminotransferase ratio for analysis.), the abbreviation ALT/AST was not written.

Author’s response: Thank you for your kind review. As your recommendation, the terminology for the ratio of liver values has been renamed ALT/AST to improve terminology consistency in the abstract.

Abstract)

Background: Insulin resistance (IR) is common pathophysiology in type 2 diabetes mellitus, cardiovascular disease, and non-alcoholic fatty liver disease, thus, screening the risk for IR becomes important to prevent disease progression. We evaluated the alanine aminotransferase/aspartate aminotransferase (ALT/AST) ratio to predict IR in the general population, regardless of comorbidity.

2. Abstract- Results: Only the findings should be included in the results section. Comments should be avoided. Findings should be presented clearly and statistically in the form of group comparisons.

Author’s response: Thank you for your kind review. As you pointed out, we've concisely and clearly presented the ambiguity. In particular, we presented the statistical methods for each result and presented only objective findings.

Abstract)

Results: Based on the PCC, ALT in both men and women were positively correlated to four IR indices, including Glc, insulin, HOMA-IR, and HOMA-β, which were consistently observed in stratification analyses based on diabetes, dyslipidemia, alcohol consumption, and obesity. In multivariate linear regression, when comparing ALT levels, ALT/AST ratio exhibited better predictive performance for Glc and HOMA-β in Korean men and provided improved outcomes for all IR indices in Korean women.

 

3. Introduction: a. Why is NAFLD described as dominant rather than insulin resistance and homo-IR? NAFLD relationship is not written in the conclusion section. If NAFLD is an primary issue, this should be stated in the study title. b. Introduction and conclusion sections are unrelated.

Author’s response: We are not focusing on the presence or absence of fatty liver, but rather analyzing the ALT/AST ratio, which is one of the factors that reflects fatty liver. We've reviewed the introduction and conclusion sections again, and as you commented, we've come to recognize that there are elements that can cause confusion. We minimized the content on NAFLD and enhanced the content of the IR for its relevance to chronic disease (DM and CVD) and as a predictor of metabolic disorders (MetS and NAFLD) as follows.

1st paragraph in Introduction

Since the prevalence of hypertension, dyslipidemia, type 2 diabetes mellitus (DM), and Non-alcoholic liver disease (NAFLD) is increasing, it is important to early detect the presence of insulin resistance (IR) without specific symptom. IR is well known risk factor for critical cardiovascular disease (CVD) outcomes. In peoples without DM, IR represented by HOMA-IR was independent risk factor for stroke and coronary vascular disease.[1-3] Even in normal body weight peoples with insulin resistance, risk of incident DM and CVD events increased compared with a reference category.[4] NAFLD, DM and metabolic syndrome shares IR as the underlying pathophysiology [5-10]. Thus, detection of IR is helpful in predicting the risk or progression of metabolic syndrome.[11, 12] According to the previous reported study, estimated liver fat content significantly increased during follow up in IR, ordinary body weight people.[4] Likewise, NAFLD without DM patients was reported that they had reduced glucose disposal metabolism likely DM patients.[13]

4. Introduction: c. alanine aminotransaminase protein (ALT). protein ?

Author’s response: Sorry for the confusion. We deleted the “protein” term.

 

5. Dataset and Participants. The patient population should be clearly differentiated. This section should be summarized. If NAFLD is present, it should be stated whether USG scanning was performed or not.

Author’s response: We are sorry that the mistakes we made in writing our paper may have prevented researchers from understanding the hypotheses of this study. We did not study the association between IR and NAFLD, but rather the relationship between IR and liver levels, a key marker of fatty liver. The KNHANES we analyzed did not include radiologic data such as US or Computed tomography. As you pointed out, the mention of NAFLD in Dataset and Participants was revised into liver profiles. 

1st paragraph in Dataset and Participants

This study analyzed the Korea National Health and Nutrition Examination Survey (KNHANES) to confirm the association between liver profiles and IR.

6. Diabetes mellitus was defined as serum fasting glucose level ≥126 mg/dL; previous diagnosis; anti-glycemic drugs. Has this been confirmed 2 times?

Author’s response: Thank you for your professional comment. In diagnosing diabetes, if the suspect is asymptomatic, it is accurate to measure fasting blood glucose once and then look at the two results to make a diagnosis. However, KNHANES only includes a single-point fasting glucose level, and we categorized diabetics based on a single-point result. We addressed the aforementioned contents as a limitation in the discussion section.

8th paragraph in Discussion

Last, deu to the design of retrospective, cross-sectional study, minor weak points occurred. For example, in diagnosing diabetes, if the suspect is asymptomatic, it is accurate to measure fasting blood glucose once and then look at the two results to make a diagnosis. However, KNHANES only includes a single-point fasting glucose level, therefore DM was defined based on a single-point result.

7. (high- and low-density lipoprotein (HDL and LDL) cholesterol.) HDL- cholesterol? LDL-cholesterol?

Author’s response: To minimize the confusion, we modified it into “high-density lipoprotein cholesterol (HDL-C) and low-density lipoprotein cholesterol (LDL-C)”.

8. Results: ALT-to-AST ratio. ALT/ALT. please make a standard abbreviation.

Author’s response: We double-checked the paper in detail and corrected all "ALT to AST ratio" terms to "ALT/AST".

9. Please write the long version of the abbreviations in table 1 in the description.

Author’s response: We changed the title of Table 1 to a title using the long version of the abbreviation.

Table 1 in Manuscript

10. What are the criteria for T1-2-3 groups.

Author’s response: ALT/AST was sorted in ascending order, then divided subjects into tertile groups. We describe in detail in the table footnote how we categorized the tertiles in Table 1.

Table 1 in Manuscript

 

11. What are the r and p values for the ALT/AST correlation with HOMA-IR?

Author’s response: We apologize for the lack of detail and the difficulty in understanding. Our Figure 1 in the main text indicates the correlation of ALT/AST with HOMA-IR and its components. The “r” was calculated by Pearson correlation coefficient (PCC) and its matched p-value was illustrated with star mark. We added a more detailed explanation to the Figure 1 as shown below.

Legend of Figure 1

12. Discussion: Findings should be discussed clearly.

Author’s response: We appreciate the comment that can improve this study. Based on your comments, in the first paragraph of the Discussion section, we summarize the main findings of our study. In subsequent paragraphs, we interpret our findings in relation to those of other studies as follows.

1st paragraph in Discussion section

Using the Korean nationwide datasets, we demonstrated the relationships of ALT and ALT/AST ratio with four IR indices. The positive association between liver profile and IR was common in both Korean men and women population. Moreover, the link between hepatic steatosis indices and IR remained independently significant after adjusting for various clinical variables. Through stratification analyses in various settings, it was observed that overall, the ALT/AST ratio was more reflective of IR status than ALT.

4th paragraph in Discussion section

HOMA-IR means the presence of insulin resistance and is one of predictor for diabetes. However, to calculate HOMA-IR, additional sampling for blood and fasting time are required. Therefore, an effective approach is needed to for screening method. There are some studies that the ALT/AST ratio could be a tool for prediction insulin resistance [38]. Kawamoto R, etc. also reported the ALT/AST ratio was the best surrogate marker for insulin resistance compared to other marker such as TG, GGT, ALT or AST alone [39]. This was also supported by the other studies [40]. Recently, the relationship between ALT/AST ratio and diabetes, pre-diabetes stage was also reported in china [41]. This study is the first report to include more than 10,000 subjects in Korea. Moreover, the predictive power was better in middle-aged women and people who with alcohol consumption (Fig 4). Based this results, in the group corresponding to T2 or T3, the prevalence of diabetes, hypertension, and HOMA-IR significantly were increased. It means the management for metabolic syndrome is required in T2, T3 group.

Answers to comments of Reviewer #2:

1. 1-İn the Method/data set and participants. Finally, we selected a population of 11,547 individuals (men: 5,001, women: 6,546) to determine the association between ALT/AST ratio and NAFLD. However, to determine the association of ALT/AST ratio with NAFLD, there must be a definite indication of the presence of NAFLD in the participants, such as abdominal ultrasonography or liver biopsy, and the association must be analysed accordingly. Otherwise, such a result cannot be reported.

Author’s response: We appreciate the Reviewer’s constructive comments. There was a mistake for finishing and trimming the last sentence. We focus the insulin resistance for general population, not for NAFLD. We correct the wrong sentence as follows.

2nd paragraph in Dataset and Participants

Finally, we selected a population of 11,547 subjects (men: 5,001, women: 6,546) to identify the association between the ALT/AST ratio and IR.

2. While the relationship with diabetes, dyslipidaemia and obesity is defined in Figure 1, was the relationship with the healthy control group analysed?

Author’s response: We apologize for the difficulty in understanding the lack of detail. The Figure 1 does not indicate the relationship between IR and the binomial distribution of disease status, but the association between each liver profile and IR. Moreover, the associational analysis was iterated using different combinations of subjects. We added the detailed figure legend as follows.

Figure 1.

3. İn the Table 1: ALT/AST ratio analysed according to tertile groups; the definition of tertile groups should be better explained. Table 1: ALT/AST ratio analysed by tertile groups; tertile groups need to be better explained. This situation prevents the evaluation of the relationship between ALT/AST ratio and insulin resistance independently of liver diseases.

Author’s response: Thank you for the helpful comments that improve this paper. Per your comment, we've added more detailed footnote to Table 1.

Table 1 in Manuscript

4. There are many studies investigating the relationship between ALT/AST ratio and insulin resistance in obese, healthy individuals, NAFLD, etc. in the literature. At this point, I recommend hypothesising by carefully reading the literature in the selection of method material.

Author’s response: Thank you for kindly recommendation about this report. The results of studies from individual laboratory or specific local populations may be somewhat biased and not generalizable. To overcome these limitations, it is necessary to validate similar theories in different populations. To highlight the strength and novelty of our study, we revised this manuscript title as “Effectiveness of ALT/AST ratio for predicting insulin resistance in Korean population: Large-scale, cross sectional cohort study”, considering your comments. There are similar results for ALT/AST ratio as metabolic markers. We included the large number of patients, and this would be the first published paper in Korean people.  

Answers to comments of Reviewer #3:

1. The type of study is better to be mentioned in the title.

Author’s response: I appreciate your kind comment. We revised the title of manuscript according your comment. We modified this manuscript title as “Effectiveness of ALT/AST ratio for predicting insulin resistance in Korean population: Large-scale, cross sectional cohort study”, considering your comments.

2. Please clarify statistical analysis section in more detail. For example, method of controlling confounders in this study.

Author’s response: Based on your comment, we have added a detailed description of multivariate analysis to the statistics section of the main text as follows.

1st paragraph in Statistics

A univariate (Model 1) or multivariate linear (Model 2 – 4) regression (LiR) models were implemented to identify the association between liver profiles (i.e., ALT, AST, and ALT/AST ratio) and index for IR (i.e., Glc, insulin, HOMA-IR, HOMA-β). In detail, we set the liver profiles and each index for IR as dependent and independent variables, respectively. Then, three different multivariate models (Model 2 – 4) were utilized to identify the independent relationship between liver profiles and IR indices according to four combinations of confounders, such as model 2 (age), model 3 (age, DM, dyslipidemia, and alcohol consumption), model 4 (age, DM, dyslipidemia, alcohol, BMI, and WC).

3. Authors need to add the strengths of this study before the paragraph explaining study limitations.

Author’s response: Thank you for your thoughtful comment. We add the strengths of our study in the discussion section. 

7th paragraph in Discussion

Our study had several strengths. First, this study includes the large population number in Korea. A recent study validated the association between ALT/AST and IR in a Japanese female population of approximately 460 people. However, our study demonstrated an association between liver levels and IR in more than 10,000 Korean men and women, which may be more generalizable [51]. Second, various analyses, including stratification analysis and predictive value, confirmed the independent relationship between ALT/AST and IR. While the prior research is similar to ours in comparing the relevance of various IR indices to ALT/AST [51], our series of analysis allows us to explore this relationship from a broader perspective and reveals a novel finding that ALT/AST has better predictive power than ALT.

4. Please check the concordance of the tenses used throughout the text.

Author’s response: We double-checked the manuscript to ensure consistency in the tenses.

5. Please define all nonstandard abbreviations used in the text and abstract.

Author’s response: We've made improvemen

---

## [Editor Report · Decision Letter 1]

11 Mar 2024

PONE-D-23-35138R1Effectiveness of ALT/AST ratio for predicting insulin resistance in Korean population: Large-scale, cross-sectional cohort studyPLOS ONE

Dear Dr. Kim,

Thank you for submitting your manuscript to PLOS ONE. After careful consideration, we feel that it has merit but does not fully meet PLOS ONE’s publication criteria as it currently stands. Therefore, we invite you to submit a revised version of the manuscript that addresses the points raised during the review process.

Please include the following items when submitting your revised manuscript:A rebuttal letter that responds to each point raised by the academic editor and reviewer(s). You should upload this letter as a separate file labeled 'Response to Reviewers'.A marked-up copy of your manuscript that highlights changes made to the original version. You should upload this as a separate file labeled 'Revised Manuscript with Track Changes'.An unmarked version of your revised paper without tracked changes. You should upload this as a separate file labeled 'Manuscript'.If applicable, we recommend that you deposit your laboratory protocols in protocols.io to enhance the reproducibility of your results. Protocols.io assigns your protocol its own identifier (DOI) so that it can be cited independently in the future. For instructions see: https://journals.plos.org/plosone/s/submission-guidelines#loc-laboratory-protocols. Additionally, PLOS ONE offers an option for publishing peer-reviewed Lab Protocol articles, which describe protocols hosted on protocols.io. Read more information on sharing protocols at https://plos.org/protocols?utm_medium=editorial-email&utm_source=authorletters&utm_campaign=protocols.

We look forward to receiving your revised manuscript.

Kind regards,

Anna Di Sessa, PhD, MD

Academic Editor

PLOS ONE

Journal Requirements:

Additional Editor Comments:

Authors have well-addressed the previous issues raised by reviewers. However, prior to be considered for publication, there are still some minor concerns:

- introduction: this section contains some repeated sentences (e.g.lines 68, 69,73). Overall, it needs to be rephrased to improve clarity.

- secondly instead of second (line 319)

- lastly instead of last (line 332)

- discussion: line 332-335, this sentence should be rephrased in a clearer manner using a scientific language. More, potential prognostic implications should be added.

- conclusion: our findings suggest ... instead of we suggest; a practical message for clinicians should be added.

- English language should be polished throughout the manuscript.

---

## [Author Response · Author response to Decision Letter 1]

8 Apr 2024

We would like to thank your kind consideration about our manuscript. We used to many words to convey our idea. So we changed the repeated sentences and words according to your recommendation, and we also received the English correction service.

---

## [Editor Report · Decision Letter 2]

24 Apr 2024

Effectiveness of the ALT/AST ratio for predicting insulin resistance in a Korean population: A large-scale, cross-sectional cohort study

PONE-D-23-35138R2

Dear Dr. Kim,

We’re pleased to inform you that your manuscript has been judged scientifically suitable for publication and will be formally accepted for publication once it meets all outstanding technical requirements.

Kind regards,

Anna Di Sessa, PhD, MD

Academic Editor

PLOS ONE
---

## [Editor Report · Acceptance letter]

7 May 2024

PONE-D-23-35138R2 

PLOS ONE

Dear Dr. Kim, 

I'm pleased to inform you that your manuscript has been deemed suitable for publication in PLOS ONE. Congratulations! Your manuscript is now being handed over to our production team.

Kind regards, 

on behalf of

Dr. Anna Di Sessa 

Academic Editor

PLOS ONE